# OpenReview forum: "Arondight: Red Teaming Large Vision Language Models with Auto-generated Multi-modal Jailbreak Prompts"
_acmmm.org/ACMMM/2024/Conference — MM2024 Poster_

### Official Review · Reviewer_rkfy · 2024-05-25

**Rating:** 5
**Confidence:** 3

**Summary:**

The authors introduce Arondight, a novel red team framework aimed at evaluating the security vulnerabilities of Vision Language Models (VLMs) in generating harmful multimodal content. A key contribution is an automated multimodal jailbreak attack that leverages reinforcement learning to optimize text prompts for diversity and integrates specially crafted toxic images to boost the effectiveness of textual attacks against VLM safety defenses.

**Strengths:**

- The work introduces the a standardized red teaming framework (Arondight) specifically designed to evaluate the security vulnerabilities of Vision Language Models (VLMs) against generating harmful multimodal content. This fills an important gap, as existing red teaming frameworks are tailored for text-only large language models.
- Arondight addresses critical challenges like modal coverage (handling both text and visual inputs) and output diversity when adapting red teaming methods from LLMs to VLMs. It integrates techniques like entropy bonuses and novelty reward metrics to incentivize diverse test case generation.
- A core innovative component is an automated multimodal jailbreak attack that leverages reinforcement learning to optimize text prompts for diversity, while also incorporating specially crafted toxic images to boost the effectiveness against VLM safety defenses.
- The paper provides a thorough evaluation across 10 cutting-edge VLMs, exposing significant vulnerabilities.

**Limitations:**

- What concerns me is the reliability of the toxicity score calculation from the textual responses. In particular, did the authors consider taking any human judgement into account while grading the toxicity scores?

**Suitability:**

3

---

### Official Review · Reviewer_L4hV · 2024-05-26

**Rating:** 6
**Confidence:** 3

**Summary:**

The paper introduces Arondight, a red team testing framework specifically designed for LVLMs. Arondight aims to evaluate the security and reliability of LVLMs by providing a standardized framework. It integrates an auto-generated multi-modal jailbreak attack strategy to create test prompts and assess the responses of LVLMs. The framework was evaluated on ten LVLMs, revealing significant security vulnerabilities in generating toxic content and aligning multi-modal prompts.

**Strengths:**

1. The authors have provided a comprehensive LVLMs security evaluation framework, enabling researchers to better understand the security of the models. The experimentation and selection of tested models are also quite exhaustive.
2. The writing is thorough, the chapter organization is well-structured, and the experimental setup is comprehensive, making it a solid piece of work.

**Limitations:**

1. The innovation of the entire framework is somewhat limited, as it appears to be quite procedural in comparison.
2. One question I would like to ask is, when you enter an attack prompt or image, especially for some commercial models such as GPT4, it will intercept these toxic prompts and tell you "I am an AI, I can't do anything harmful to human beings" or something like that. How do you handle this situation?

**Suitability:**

3

---

### Official Review · Reviewer_1LqW · 2024-05-27

**Rating:** 3
**Confidence:** 2

**Summary:**

The paper introduces a novel framework designed to evaluate the security vulnerabilities of Large Vision Language Models (VLMs). They proposed as a red team framework specifically tailored for VLMs, focusing on the visual modality and diversity issues not sufficiently covered by existing methodologies.

Arondight incorporates an automated multi-modal jailbreak attack, using a red team VLM to produce visual prompts and a red team LLM guided by a reinforcement learning agent to generate textual prompts. The paper reports an average attack success rate of 84.5% on GPT-4 across all fourteen prohibited scenarios defined by OpenAI for generating toxic text.

**Strengths:**

1. The paper presents a pioneering framework for evaluating VLMs, addressing a significant gap in the current security assessment methodologies for these models.

2. Arondight's multi-modal approach to generating jailbreak prompts ensures a more thorough testing of VLMs against a wider array of potential threats.

3. The reported 84.5% success rate in jailbreak attacks against GPT-4 is impressive and underscores the framework's effectiveness.

**Limitations:**

1. Does the Arondight framework's reliance on the target VLM's security mechanisms to identify inappropriate responses mean that if these mechanisms are not robust or can be easily bypassed, the framework might fail to adequately assess the VLM's security against more advanced attacks?

2. The paper mentions using entropy rewards and novelty rewards to increase the diversity of test cases, but if the dataset used to train the RL agent has biases or limitations, could this lead to the generated test cases failing to cover all possible security scenarios, thereby affecting the comprehensiveness of the evaluation?

3. The Arondight framework utilizes adversarial images and text to enhance jailbreak attacks. However, the generation of adversarial samples might require a deep understanding of the VLM's security defense strategies. If the VLMs update their security mechanisms, could previously effective adversarial samples become ineffective?

4. Despite the paper emphasizing adherence to strict ethical guidelines and promptly reporting jailbreak techniques to service providers, could the generation of harmful content in test cases raise ethical and legal issues, how to ensure rigorous regulation?

5. The paper does not discuss the long-term effectiveness of the framework or how it might need to evolve with advancing VLM technologies.

6. Some sections of the paper could benefit from more detailed explanations, particularly for readers who may not be familiar with red teaming or reinforcement learning.

**Suitability:**

3

---

### Meta-Review · Area_Chair_CnWr · 2024-07-03

**Recommendation:** Accept (Poster)
**Confidence:** 4

**Metareview:**

The paper demonstrates several strengths, including an innovative framework for evaluating Vision Language Models (VLMs), a comprehensive evaluation across multiple models, and clear presentation of the findings. However, notable weaknesses include insufficient discussion on the framework's limitations for advanced VLM scenarios and potential biases in training data. In the rebuttal, the authors adequately addressed some reviewer concerns. Final ratings from reviewers varied between borderline reject, accept, and weak accept. Despite a negative recommendation from one reviewer, suggesting more in-depth discussions, the Associate Chair (AC) recommends accepting the paper.

---

### Meta-Review · Senior_Area_Chairs · 2024-07-10

**Recommendation:** Accept (Poster)
**Confidence:** 4

**Metareview:**

This paper received mixed rating initially. After rebuttal, there is still a BA  to question the in-depth discussions. SAC and AC carefully checked the paper, reviews and rebuttal and recommend acceptance of the paper.